# A Novel Textile Stitch-Based Strain Sensor for Wearable End Users

**DOI:** 10.3390/ma12091469

**Published:** 2019-05-07

**Authors:** Orathai Tangsirinaruenart, George Stylios

**Affiliations:** Research Institute for Flexible Materials, Heriot Watt University, Edinburgh EH14 4AS, UK; ot32@hw.ac.uk

**Keywords:** textile-based stretch sensors, stitch structure, wearable stretch sensor, conductive thread

## Abstract

This research presents an investigation of novel textile-based strain sensors and evaluates their performance. The electrical resistance and mechanical properties of seven different textile sensors were measured. The sensors are made up of a conductive thread, composed of silver plated nylon 117/17 2-ply, 33 tex and 234/34 4-ply, 92 tex and formed in different stitch structures (304, 406, 506, 605), and sewn directly onto a knit fabric substrate (4.44 tex/2 ply, with 2.22, 4.44 and 7.78 tex spandex and 7.78 tex/2 ply, with 2.22 and 4.44 tex spandex). Analysis of the effects of elongation with respect to resistance indicated the ideal configuration for electrical properties, especially electrical sensitivity and repeatability. The optimum linear working range of the sensor with minimal hysteresis was found, and the sensor’s gauge factor indicated that the sensitivity of the sensor varied significantly with repeating cycles. The electrical resistance of the various stitch structures changed significantly, while the amount of drift remained negligible. Stitch 304 2-ply was found to be the most suitable for strain movement. This sensor has a wide working range, well past 50%, and linearity (R^2^ is 0.984), low hysteresis (6.25% ΔR), good gauge factor (1.61), and baseline resistance (125 Ω), as well as good repeatability (drift in R^2^ is −0.0073). The stitch-based sensor developed in this research is expected to find applications in garments as wearables for physiological wellbeing monitoring such as body movement, heart monitoring, and limb articulation measurement.

## 1. Introduction

In the last decade there has been an increasing interest for developing different types of wearable sensors. There are many sensor types that have shown potential as wearable sensors amongst them piezoresistive films which show good change in resistance by simple changes to their geometry and observed microcracking contributing to high gauge factors [1,2]; although their durability and strechability are areas of concern for further development. Capacitive sensors are another common type seeing in touch screens because of their good sensitivity, low energy and adaptability [3]. They have however mainly being used for pressure because they suffer from environmental noise and hence difficult for use in wearable applications. Textile-based sensors are desirable for wearable end users [4,5] because they are comfortable, flexible, and not obstructive to the wearer’s everyday activities. A textile sensor can be designed and presented in numerous types and forms. One particularly interesting type is as a strain resistance sensor, which is achieved by the alteration of the mechanical properties of the material under stress/strain deformation, whilst its flexibility allow care of wrapping of the body of the wearer. Textile strain sensors can detect stretch, displacement, and force resulting from large movement of joint bending [6] or small body movements such as breathing [7]. Many studies have investigated theoretical and practical relationships between the electrical resistance and elongation of conductive fabrics [8,9,10]. Thomson and Kelvin [11] found that the resistance changes of a conductor were affected by stretching. Zhang et al. [12,13] and Li et al. [14] showed that the electrical resistance occurring between overlapping knitted loops is a major factor contributing to the overall resistance. Likewise, Ehrmann et al. [15] found that different yarns, stitch dimensions, and different fabric directions affect the elongation and time-dependent resistance behavior of the sensor.

There are many changes in properties when a knitted fabric-based stretch sensor is extended. Previous researches [16,17,18] have shown that one interesting effect of stretching a fabric is that it increases its electrical resistance along the conductive thread. This is caused by opening up the stitch and so breaking the parallel contact points, making it necessary for the current to travel in series rather than in parallel. This increase in conductive path leads to greater resistance [17,19,20] as shown in Figure 1.

Another effect is due to the stretching of the conductive yarn itself [20]. It is well known that when a conductive material is stretched its cross-section is reduced, while its conductive path is increased, leading to greater resistance. The relative dominance of each effect is dependent on the stitch type, the strain and the type of conductive yarn.

In this study, our objective is to explore the prospect of developing a flexible wearable textile strain sensor. The optimum design of such sensor is investigated in different types of stitch structures, having as criteria the ability to measure strain deformations and electromechanical properties. Hence, a study was under taken in which conductive threads were sewn under different configurations, directly onto a knitted fabric and their characteristics investigated. 

## 2. Experimental Materials

In this study, our primary aim is to find a particular stitch assembly that has a good stretch ability as well as electrical performance. The stitch assembly is formed by using conductive thread and is constructed using stretchable single jersey fabrics which allow the stitches to be flexible, unobtrusive, and achieve linearity. The preliminary studies of fabric suitability were undertaken to find the ideal base fabric. Six different fabrics were investigated under the criteria of having high elastic recovery, Table 1. A single jersey of Nylon 4.44 tex/2-ply with Spandex 7.78 tex and weight of 260 g/m^2^ was found ideal for use as the fabric substrate base sensor, because it has the highest elastic recovery at 93%, as shown in Figure 2.

Trials with different stitching configurations followed. Seven samples 50 × 250 mm^2^ were made by stitching along the wale direction of the fabric. Samples with four stitch types—304 (Zigzag stitch), 406 (2-needle multithread chain stitch: rear side), 506 (4 threads overlock stitch), and 605 (3-needles covering chain stitch)—were constructed. Two types of conductive threads used were silver plated nylon 117/17 2-ply and 235/34 4-ply, Statex Productions & Vertriebs GmbH, Bremen, Germany. Their electrical specification is given in Table 2, and their properties in Table 3. Both threads are commercial, and they have also been used by other researchers [16,17,21,22,23,24].

These conductive sewing threads are lightweight, flexible, soft, durable, strong, and do not suffer from permanent distortion after being bent. Therefore they are ideal for machine sewing for typical garment operations. The electrical resistance of the two threads used are given in Figure 3. After imposing the threads to five-cycle tensile testing shown in Figure 4, the resistance response of both threads increases as they are stretched (loading) and decreases upon relaxation (unloading). This good overall electrical behavior renders them suitable for wearable end-uses. The results also show that the threads have a small delay in becoming fully relaxed.

For testing consistency, three of the four stitch types have the same characteristics (5 stitches per cm and 7 mm wide). However, stitch type 304 by virtue of its design needs to have different dimensions: 11 stitches per cm and 3 mm wide, as shown in Table 4 where the conductive thread is represented by the black line in the stitch structure.

### 2.1. Experimental Methods

#### 2.1.1. Measured Properties and Characteristics

In order to investigate the suitability of the different stitch types, various properties and characteristics of each stitch were measured, which will briefly be defined here with reference to Figure 5.

##### Working Range

The working range of a sensor determines its end-use suitability. A sensor with a wider working range will be more adaptable to a wider variety of uses. The working range is measured as a percentage of strain, that being the change in length divided by (the unstretched original) length. A sensor which has a working range starting from its unstretched condition (rest position, i.e., zero) will be more useful than a sensor with a working range between two stretched states. It is clear that the sensor in Figure 5 has a working range from 0 to 15%, but above that it becomes unsuitable due to nonlinearity and nonmonotonicity.

##### Monotonicity

A sensor will be more useful if it is monotonic. This means that the resistance changes in a constant incremental value upward or downward as the strain increases or decreases, respectively. The resistance value of the sensor in Figure 5 does not alter in its constant increment; instead it is reaching to a peak at 20% strain and then falling immediately after. This means that, when in use, it would be difficult to determine if the sensor is giving a measurement of resistance changes of say 0.25% because its strain can be in the 8 to 12% or in 35 to 45% region. So we say that this sensor has no monotonicity.

##### Gauge Factor

The sensitivity of sensor resistance changes to strain, is given by
$$Gauge\;Factor=\frac{\left(\frac{\Delta R}{R}\right)}{\left(\frac{\Delta l}{l}\right)}=\frac{\left(\frac{\Delta R}{R}\right)}{\varepsilon }$$
where R and l are the unstretched resistance and length, respectively; ΔR and Δl are the changes in resistance and length due to stretching, respectively; and ε is the strain. The gauge factor is unitless and is calculated by taking the gradient of lines of best fit on a graph of resistance change vs. strain. For example, the line of best fit in Figure 5 follows equation y = 0.0025x + 0.2065, so its gauge factor is 0.0025. The ideal sensor would have a high gauge factor (high sensitivity), so that changes in resistance are large in relation to strain changes. This would make it easy to detect a small change in strain because these would show as a large change in resistance. If the sensor shows very small changes in resistance in relation to changes in strain (i.e., a resistance vs. strain graph showing a horizontal strain line), it makes it very difficult to accurately measure small changes in strain, hence leading to poor sensor performance.

##### Linearity

Linearity is the proportion of change in resistance in relation to the proportion of change in strain. A perfect sensor would have a high and uniform resistance change over the whole of the strain range, so making it easy to calculate the strain from any given resistance measurement. There are various metrics for linearity. The metrics used here are visual inspection of the graphs along with assessment of the R^2^ value (coefficient of determination), which indicates how well a line of best fit represents the data it is fitted to. A straight line cannot fit well to a highly curved relationship between resistance and strain, so giving low R^2^ values, as shown in Figure 5. It should be noted that low R^2^ values could also be caused by random data scatter or large hysteresis.

##### Hysteresis

The hysteresis of a sensor is the difference between the resistance at any given strain on the loading cycle and the resistance at that same strain on the unloading cycle. Ideally, the sensor would have the same resistance at that strain point regardless of whether it is loading or unloading, so that one strain value can always be measured with one resistance value, without the need to know which cycle the sensor is in, which is impractical. Figure 5 shows a high hysteresis. If a resistance change of say 0.2% was measured, it could correspond to a strain of 5% or 10%, depending on the direction of loading.

##### Repeatability

Drift is another characteristic, which is determined by repeating cycles, measured by any changing characteristic divided by the number of cycles. In this work, the change of the unstretched resistance, and the change in the gauge factor, between the second and 99th cycles (for reasons explained later) was taken as an indicator of repeatability.

##### Other Properties/Characteristics Not Investigated

There are many other factors that determine the suitability of a wearable sensor and its end-use, including response/time delay, creep, sensitivity to other types of deformations (e.g., twisting), sensitivity to environmental conditions, stiffness, ease of manufacture, comfort, and aesthetics. Although these other properties may be important they are not essential for the purpose of this investigation.

### 2.2. Experimental Apparatus and Procedure

Figure 6 shows the experimental apparatus and set up. The sensor samples are clamped between the jaws of an Instron 3345 Tensile Tester and the two ends of the sensor are connected to a digital millimeter (DMM Agilent U1273A/U1273AX, Agilent Technologies Inc., Santa Clara, CA, USA). Installing the Agilent GUI Data Logger software on a PC is used to record the electrical resistance response to extension and recovery cycles, at a sampling rate of approximately 1 Hz. The jaws were electrically isolated from the fabric sensor with a layer of synthetic polymer rubber, so that only the resistance of the sensor itself would be measured by the millimeter. The jaws were set 150 mm apart and 250 mm long in the sensor direction (50 mm is needed at either side of the sample for clamping).

Data from the tensile tester and millimeter were subsequently aligned and overlapped using digital timestamps. Each cyclic test is performed at the rate of 200 mm/min and at 50% extension in 10 cycles. In order to observe the repeatability of the samples further tests of 100 cycles were also performed.

Standard atmospheric conditions were used for the experiment i.e.; temperature at 20 ± 2 °C and 65% ± 2% R.H, and all samples were allowed 24 h in the lab for conditioning, prior to testing. 

### 2.3. Data Analysis Procedure

This section will describe and analyze the way in which the various sensor characteristics were derived from the stress/strain required data. Extension and resistance were converted into strain (%) and percentage resistance change (%), as below.
$$\varepsilon =\frac{\Delta l}{l}\;Resistance\;Change\;\left(\%\right)=\frac{\Delta R}{R}$$
where Δl is the extension value from the tensometer and l is the unstretched gauge length, Figure 6 (150 mm). Calculating the percentage resistance change was more complicated because in this study we are interested in the percentage resistance change of the stretching portion of the sensor only and not including the portion within the jaws. The portion in the jaws gives a constant additional resistance, herein called “resistance bias”. In the unstretched state, the sensor was assumed to have an approximately uniform resistance per mm length. Therefore the resistance of the clamped portion (the resistance bias), is given by
$$R_{bias}=\left(proportion\;of\;length\;in\;clamps\right)\times \left(\mathrm{total}\;\mathrm{unstretched}\;\mathrm{resistance}\right)$$

The proportion of length in the clamps is taken from Figure 6 as 0.4 (100/250 mm), while the total unstretched resistance is different for each stitch and was taken as the resistance at time = 0 (the first data sample). The resistance bias is subtracted from all resistance data samples to give the resistance of the unclamped portion at any one time. To calculate the percentage resistance change, an original or baseline resistance is needed. Figure 7 shows a typical plot of resistance vs. time.

Note that the unstretched resistance in the first cycle (circled in red) is significantly lower than that of subsequent cycles, where it becomes stable. Therefore, the first cycle of each sample is not included in the data. Hence, for end use measurements it is advisable that sensors are pre-stretched in order to normalize the data, this was also observed by others [17,19]. In our experiments from now on, only data from the second cycle onwards is being used (circled in green).

Our aim is to find the values of working range and gauge factor for each stitch type. However, there is likely to be some amount of variation between cycles, therefore data for a number of cycles were laid on top of each other and averaged to find an assembly average. Ten cycles were used starting from the second cycle, as discussed. For example, the resistance at the start of the cycle was taken to be the average of the resistances at the start of cycles 2 through to 11. The second value of resistance was the average of the second sample of resistance for cycles 2 through to 11, and so on, until a complete averaged cycle for each stitch type was established. The strain data was processed similarly. An example of the percentage resistance change vs. strain for a sample average is given in Figure 8.

The grey curve is the assembly average over 10 cycles for this particular stitch, and it is reaching approximately 25% of strain beyond where the response becomes highly nonlinear and nonmonotonic. Therefore the working range of this sensor is considered to be 0 to 25% strain. This is marked on the graph by the blue and red lines, denoting loading and unloading directions respectively. A green line of best fit was then fitted to the data within the working range, revealing the gauge factor of this sensor (in this case 3.71), and a coefficient of determination (R^2^, in this case 0.95). As stated, R^2^ is a measure of how well the data fits the line of best fit, indicating nonlinearity, scatter, and spread due to hysteresis.

Finally, to assess repeatability, the resistance change vs. strain graphs of the 2nd and 99th cycles were compared, as shown in Figure 9. The second cycle was used for reasons explained earlier, and the 99th cycle was used because the 100th cycle showed some discontinuity due to machine stopping. This graph revealed changes in characteristics such as gauge factor, hysteresis and unstretched resistance. During experiments stitch structure 304 could only be realized with 2-ply 33 tex sewing thread as the 4-ply 92 tex thread was distorting the fabric (bouncing), due to the tensions and geometry of this particular stitch type. 

## 3. Experimental Results, Analysis and Discussion

To help explain early sensor characteristics, close-up photos were also taken for visual inspection of the stitch deformation. Table 5 shows close-up photographs of one typical sample of each stitch type, relaxed, and stretched, which will be referred to as the data is analyzed further.

Figure 10 shows the percentage resistance change vs. strain for the various stitch types, averaged over all samples of each stitch type, and over the 2nd to 11th cycles. Data derived from these graphs are tabulated in Table 6.

The most striking aspect of the percentage resistance change vs. strain of the graphs is the performance of the 304 2-ply stitch. This is highly linear, monotonic, and has a working range that appears to potentially pass 50% extension. It also features low hysteresis and a reasonable high gauge factor, corroborating good linearity. Stitch 506, 4-ply also shows some good attributes, whilst all other stitches followed a trend of increasing resistance followed by decreasing resistance, as the fabric is stretched. This limited their working range to a maximum of 8 to 25% (406 4-ply and 2-ply respectively), which suggests that there may be competing effects in their resistance from extension. 

From theory previously described, it is thought that one effect of extending the fabric is to open up contacts between adjacent lengths of conductive thread, thus increasing the conductive path by moving the resistive lengths into series rather than parallel and thus increasing resistance. This might account for the rise in resistance. In all stitches, the initial sensitivity of resistance to strain is low, before increasing rapidly. The likely cause of this is that few contacts are opened up before a certain level of strain is reached. 

Something, which might account for the decrease in resistance is the change in dimensions of the conductive thread itself, is altering the resistance per unit length locally. It was previously stated that a conductor increases resistance as it is stretched, due to the longer conduction path and reduction in cross-sectional area. Although one might expect the total length of the conductive thread to increase as the fabric is stretched, it may be the case that it decreases, in certain stitch configurations, hence reducing the sensor resistance. Figure 11 shows an example of fabric stretching with a stitch geometry that might lead to a reduction in the overall conductive thread length.

Note that the main fabric is elongating as well as narrowing when it is stretched. This means that any portions of conductive thread that are in the direction of the overall stretch are also being elongated; however, those that are 90° to its length (i.e., lying across the fabric) will be shortening in length (compressing). In this stitch, there is more conductive thread length lying across the width than in the direction of stretch; therefore, overall the thread will be compressed, not stretched, hence reducing the sensor resistance.

Another explanation can be given if one examines the sewing thread tension performance when stretched on the tensile tester, shown in Figure 3, which leads to an increase in surface contact between conductive surfaces as the cross-section of the thread is expected to decrease under tension along its axis and improves in orientation along the direction of loading. Having said this, in this sensor, the sewing thread is not being stretched along its axis and hence any observation to that effect is difficult.

Another trend is that 4-ply stitches compared to their 2-ply counterparts have lower resistances. This would be expected due to the increased cross-sectional area of conducting available for the current to flow through. Stitches 506 2-ply and 605 2-ply had particularly high baseline resistances of 649 and 240 Ω, respectively. A higher baseline resistance means that, for a given gauge factor, changes in strain would produce larger changes in resistance. However, the high resistance sensors also tend to have a much lower gauge factor (percent change of resistance to strain), so offsetting this advantage.

Large differences in hysteresis were seen across the full sensor range, but when taken as a proportion of the output of the sensors, it is easier to see visually on the graphs. Within their working range, stitch type 605 showed low hysteresis, again taken as a proportion of its output, while stitch type 506 showed the highest hysteresis. Type 605 sensors also had good R^2^ values within its working range, reflecting low hysteresis and high linearity. 

Figure 12 shows graphs of percentage resistance change vs. strain of stitches between the 2nd and 99th cycles. Data derived from these graphs are tabulated in Table 7 and Table 8.

Five out of seven stitches have shown an overall increase in resistance between cycles 2 to 99. It was noted that, after stretching, the fabric does not return completely to its original length. This is because any contact between the loops of the stitch structure are reduced, hence lengthening the conductive path and increasing resistance. This means that the resistance in the relaxed state increases after each stretch cycle. Each stitch type is affected by this phenomenon to a different degree. This is because the conductive path is increased by differing amounts according to how much contact was made between loop before and after stretching.

It should be pointed out that other textile sensors may have similar gauge factor but their working range and other performance properties are very different [20].

The most uniform stitch result in the relaxed state was stitch 304 which is due to its loop contact being broken gradually. The least uniform result was stitch 605, which is due to its loop contact being broken suddenly. This phenomenon may be reduced in all sensor types by using a greater stitch density, so improving contact and making the subsequent contact breakage more gradual.

## 4. Conclusion

In this research, different types of textile stitch-based strain sensors have been investigated. The effects on their sensing properties related to their resistance change have been examined for a number of different stitch geometries. All sensors have shown significant levels of change in resistance depending on their stitch structure. It has been shown that stitch type variations have a significant effect on cyclic conductivity and resistance, revealing that each stitch design is more suitable for different sensor applications. Hence their suitability depends on the specific end-use requirements, i.e., limb articulation, heart monitoring, or respiration.

It should be pointed out that other textile sensors may have similar gauge factor but their working range and base line resistance are very different. Therefore, overall, this sensor is superior to other similar sensors that may have the same gauge factor but lack in range, linearity, and repeatability.

The sensor made with stitch type 304 and 2-ply 33 tex sewing threads was found to be the best performing in strain sensing for wearable garment end-uses, having investigated its performance in four different nylon/spandex single jersey fabrics It features an exceptionally wide working range (potentially well past 50%), good linearity (R^2^ is 0.984), low hysteresis (6.25% ΔR), good gauge factor (1.61), and baseline resistance (125 Ω), as well as good repeatability (drift in R^2^ is −0.0073). Therefore, overall, this sensor is superior to other similar sensors that may have the same gauge factor but lack in range, linearity, and repeatability.

Textile-based sensors have the ability to replace solid-state sensors and the next generation of garments will be capable of providing physiological measurement to users. 

## Figures and Tables

**Figure 1 materials-12-01469-f001:**
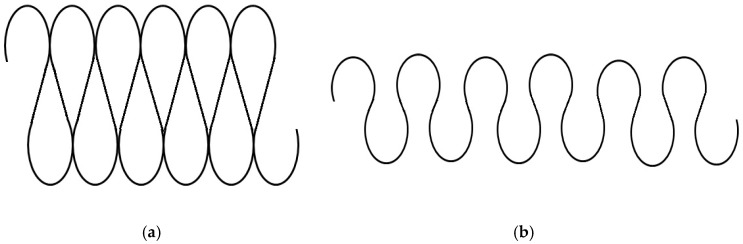
Conductive path knitted fabric model (**a**) in the relaxed and (**b**) stretched position [1,15,20].

**Figure 2 materials-12-01469-f002:**
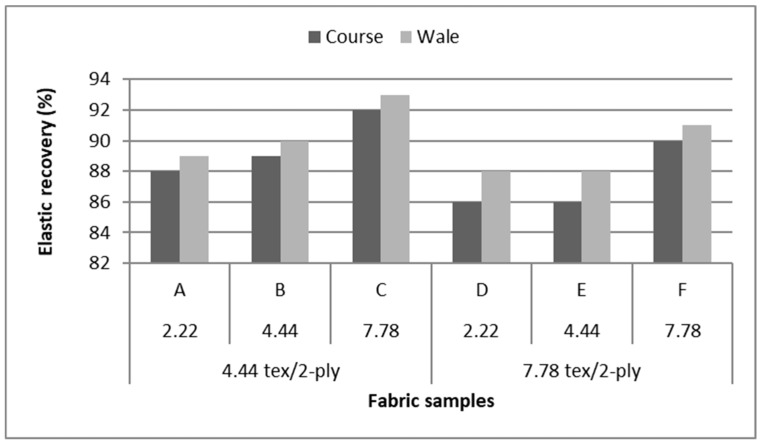
Elastic recovery values of six different nylon/spandex fabrics in course-wise and wale-wise direction.

**Figure 3 materials-12-01469-f003:**
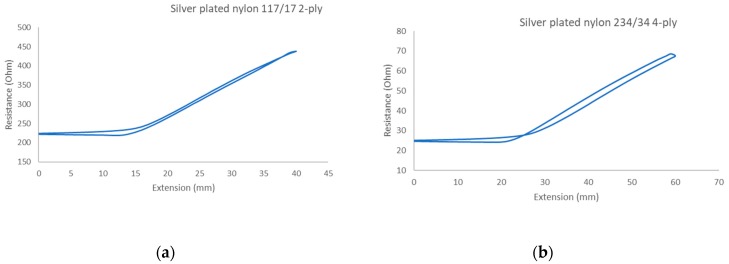
Resistance against extension and resistance against time in different type of silver plated nylon conductive thread: (**a**) 117/17 2-ply and (**b**) 234/34 4-ply.

**Figure 4 materials-12-01469-f004:**
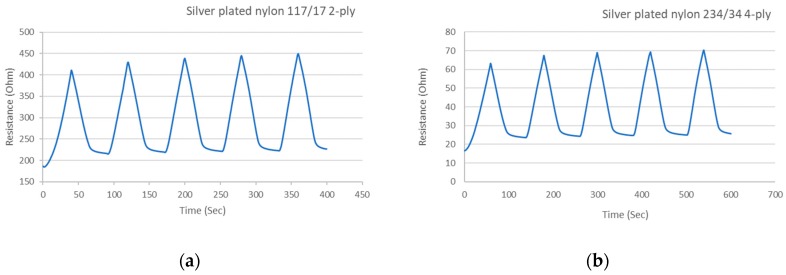
Sewing thread tensile performance during 5 cycles. (**a**) 117/17 2-ply; (**b**) 234/34 4-ply.

**Figure 5 materials-12-01469-f005:**
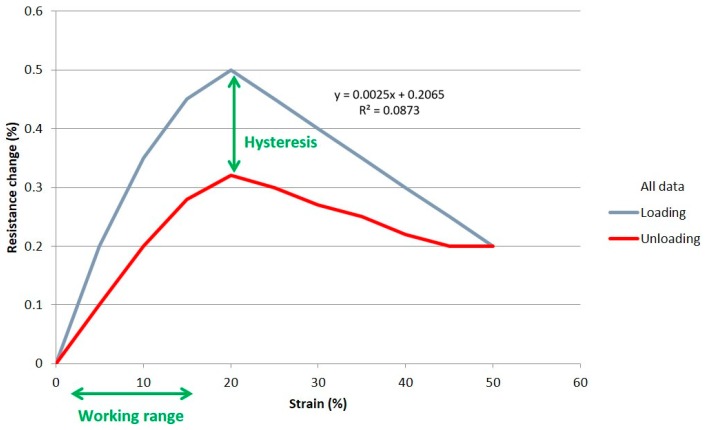
A typical graph example of resistance change (%) vs. strain (%), for defining sensor characteristics.

**Figure 6 materials-12-01469-f006:**
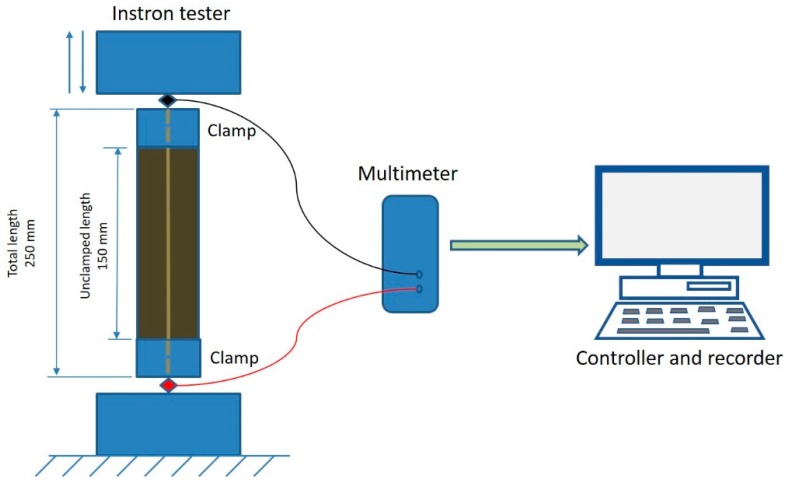
Experimental setup.

**Figure 7 materials-12-01469-f007:**
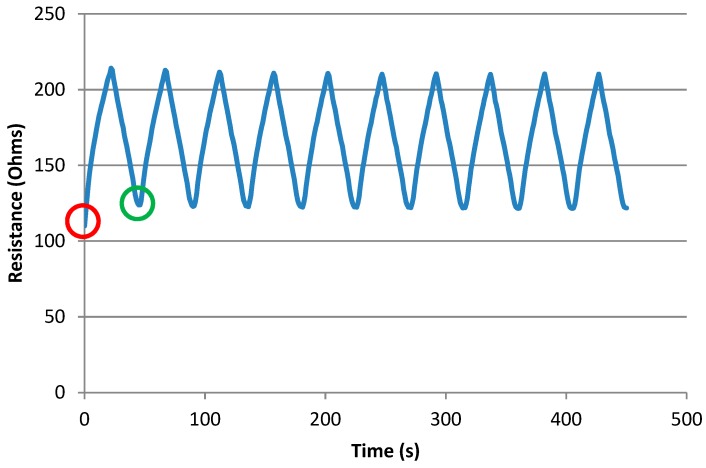
Example of resistance vs. time plot.

**Figure 8 materials-12-01469-f008:**
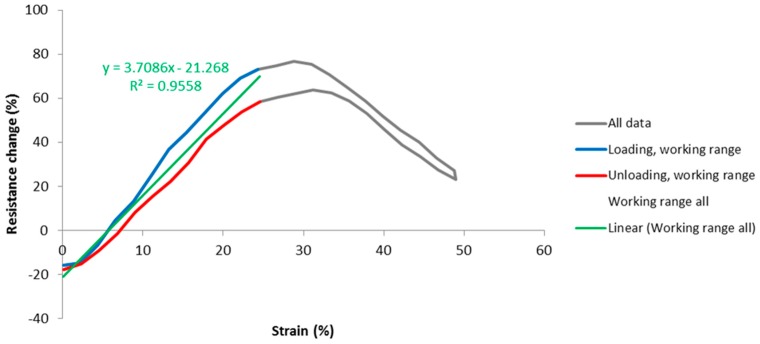
Example of assembly average over 10 cycles.

**Figure 9 materials-12-01469-f009:**
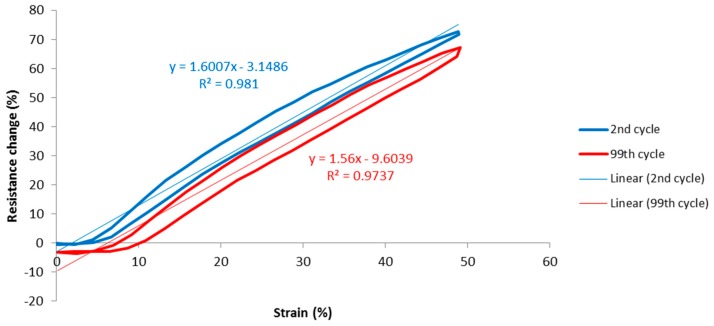
Example repeatability graph.

**Figure 10 materials-12-01469-f010:**
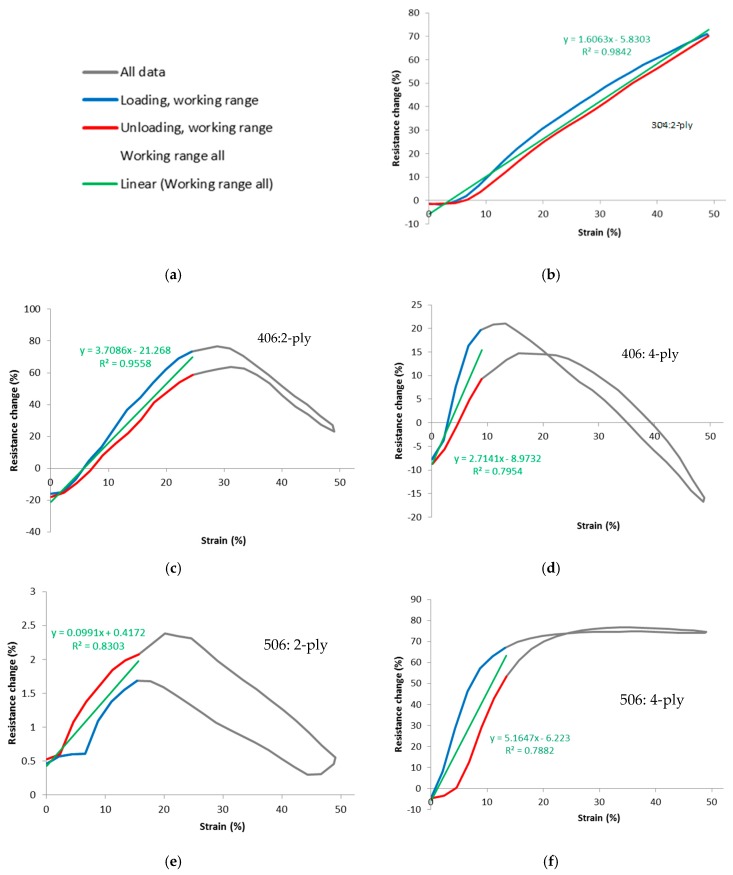
Graphs of percentage resistance change vs. strain, assembly averages over the 2nd to 11th cycles. (**a**) Legend for all graphs; (**b**) 304: 2-ply; (**c**) 406: 2-ply; (**d**) 406: 4-ply; (**e**) 506: 2-ply; (**f**) 506: 4-ply; (**g**) 605: 2-ply; and (**h**) 605: 4-ply. Note the different vertical axis scales. 2-ply is thread 117/17, 33 tex and 4-ply is thread 235/34, 92 tex.

**Figure 11 materials-12-01469-f011:**
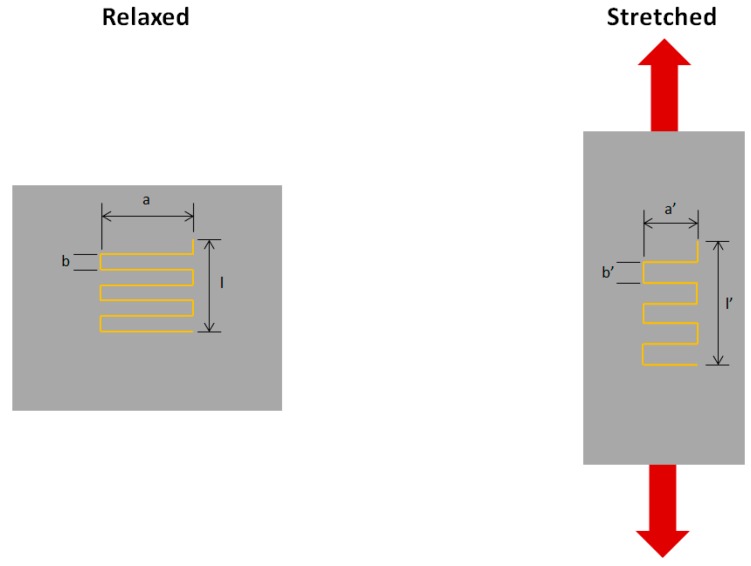
Deformation of the conductive thread.

**Figure 12 materials-12-01469-f012:**
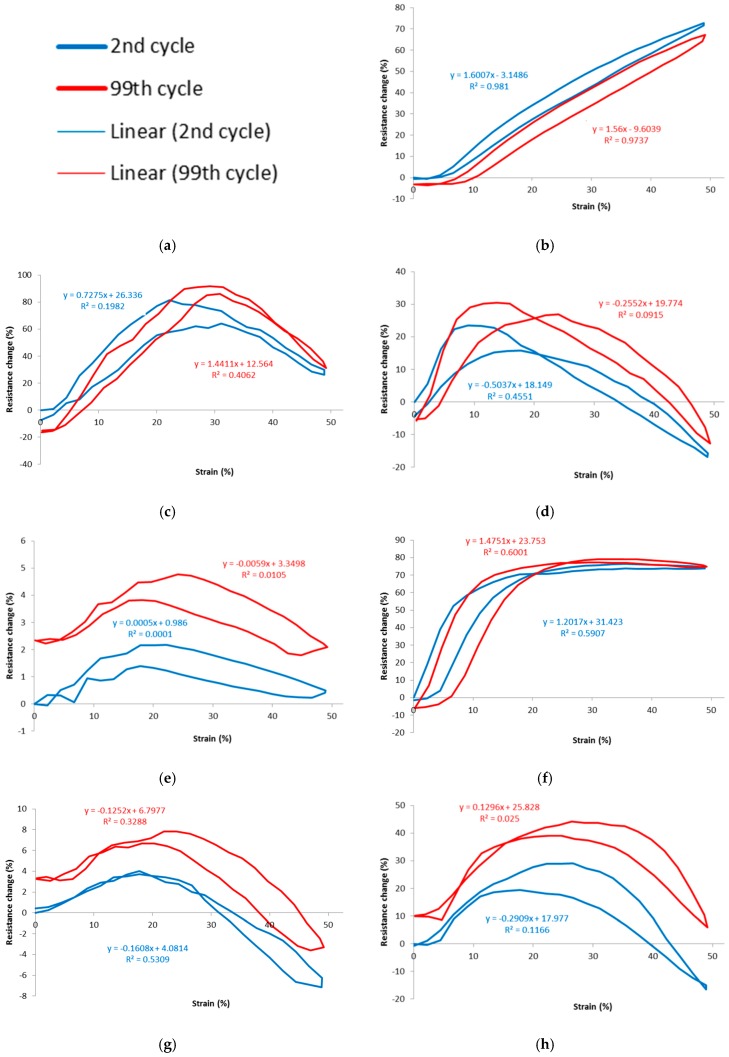
Graphs of percentage resistance change vs. strain, for the 2nd and 99th cycles. (**a**) Legend for all graphs; (**b**) 304: 2-ply; (**c**) 406: 2-ply; (**d**) 406: 4-ply; (**e**) 506: 2-ply; (**f**) 506: 4-ply; (**g**) 605: 2-ply; and (**h**) 605: 4-ply. Note the different vertical axis scales. 2-ply is thread 117/17, 33 tex and 4- ply is thread 235/34, 92 tex.

**Table 1 materials-12-01469-t001:** Specification of single jerseys fabrics.

Nylon Yarn Count (Tex)	Sample	Spandex Yarn Count (Tex)	Content Nylon/Spandex (%)	Weight g/m^2^	Thickness mm	Yarn Density per cm	
						Courses	Wales
4.44/2-ply	A	2.22	91/9	212	0.60	24	34
	B	4.44	88/12	223	0.60	23	33
	C	7.78	75/25	260	0.55	20	39
7.78/2-ply	D	2.22	94/6	286	0.65	19	33
	E	4.44	93/7	313	0.65	19	36
	F	7.78	83/17	323	0.60	18	33

**Table 2 materials-12-01469-t002:** Specification of conductive threads.

Conductive Thread	Thread Size (Tex)	Linear Resistance (Ω)
Silver plated nylon 117/17 2-ply	33	500 Ω/meter
Silver plated nylon 235/34 4-ply	92	50 Ώ/meter

**Table 3 materials-12-01469-t003:** Properties of conductive threads.

Conductive Thread	Maximum Load (N)	Energy at Break (N)	Extension at Break (mm)	Elongation (%)	Tenacity (N/Tex)
Silver plated nylon 117/17 2-ply	10.78	10.77	65.25	26.1	0.4
Silver plated nylon 234/34 4-ply	46.27	45.88	113.17	45.27	0.365

**Table 4 materials-12-01469-t004:** The experimental stitch structure sensors.

Stitch Type	Stitch Structure	Stitch per 10 mm	Stitch Width (mm)	Close-Up of the Stitched Samples
304	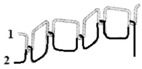	11	3	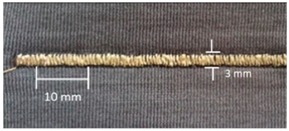
406	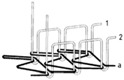	5	7	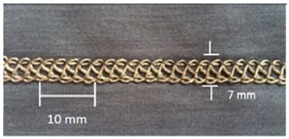
506	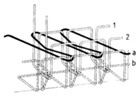	5	7	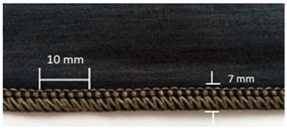
605	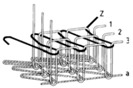	5	7	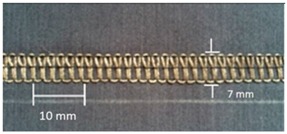

**Table 5 materials-12-01469-t005:** Close-up photographs of one sample of each stitch type, relaxed, and stretched.

Stitch Type	Conductive Thread Tex	Relaxed State	Stretched State
304	2-ply 33	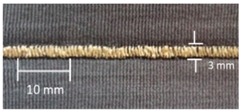	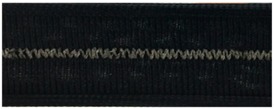
406	2-ply 33	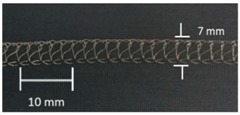	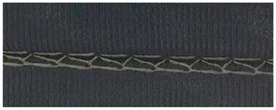
	4-ply 92	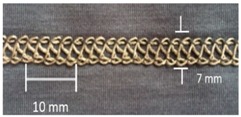	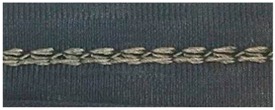
506	2-ply 33	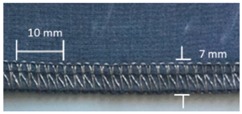	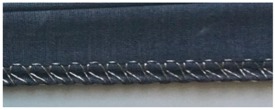
	4-ply 92	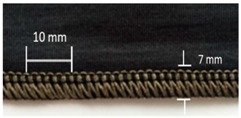	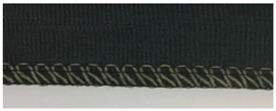
605	2-ply 33	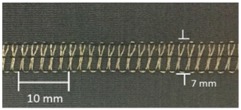	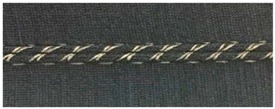
	4-ply 92	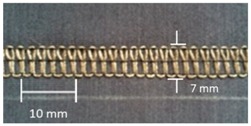	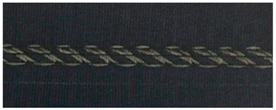

**Table 6 materials-12-01469-t006:** Tabulated data from percentage resistance change vs. strain of the assembly from graph averages. Stitch 2-ply uses 117/17 33 tex thread and 4-ply uses 235/34 92 tex.

Stitch Thread Type Tex	Working Range (% strain)	Gauge Factor Over Working Range	Baseline Resistance (Ω)	R^2^ Overall Range	R^2^ Over Working Range	Max Hysteresis Overall Range (% ΔR)
304 2-ply	0-50	1.61	125	0.984	0.984	6.25
406 2-ply	0-25	3.71	71.5	0.350	0.956	15.1
406 4-ply	0-8	2.71	55.6	0.255	0.795	11.4
506 2-ply	0-16	0.0991	649	0.0419	0.830	0.975
506 4-ply	0-12	5.16	46.7	0.601	0.788	33.6
605 2-ply	0-18	0.206	240	0.499	0.973	2.02
605 4-ply	0-18	1.65	39.3	0.0254	0.955	11.1

**Table 7 materials-12-01469-t007:** Change in gauge factor from 2nd to 99th cycles. Calculated across the entire extension. 2-ply is thread 117/17, 33 tex and 4- ply is thread 235/34, 92 tex.

Stitch Type Thread Type	G.F. 2nd Cycle	G.F. 99th Cycle	G.F. Drift	Percentage G.F. Drift (%)
304 2-ply	1.60	1.56	−0.0497	−2.54
406 2-ply	0.728	1.44	0.714	98.1
406 4-ply	−0.504	−0.255	0.249	−49.3
506 2-ply	0.0005	−0.0059	−0.0064	−1280
506 4-ply	1.20	1.48	0.273	22.8
605 2-ply	−0.161	−0.125	0.0356	−22.1
605 4-ply	−0.291	0.130	0.421	−145

**Table 8 materials-12-01469-t008:** Change in relaxed percentage resistance change and R^2^ values from 2nd to 99th cycles calculated across entire extension. 2-ply is thread 117/17, 33 tex and 4- ply is thread 235/34, 92 tex.

Stitch Type Thread Type	Relaxed ΔR 2nd Cycle (Baseline) (%)	Relaxed ΔR 99th Cycle (%)	Relaxed ΔR Drift	R^2^ 2nd Cycle	R^2^ 99th Cycle	Drift in R^2^
304 2-ply	0	−3.25	−3.25	0.981	0.974	−0.0073
406 2-ply	0	−15.2	−15.2	0.198	0.406	0.208
406 4-ply	0	−5.70	−5.70	0.451	0.0915	−0.360
506 2-ply	0	2.33	2.33	0.0001	0.0105	0.0104
506 4-ply	0	−5.68	−5.68	0.591	0.600	0.0094
605 2-ply	0	3.23	3.23	0.531	0.329	−0.202
605 4-ply	0	9.88	9.88	0.117	0.025	−0.0916

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
