# Peer review of "A Novel Textile Stitch-Based Strain Sensor for Wearable End Users"

_materials, 2019, doi:10.3390/ma12091469_

Round 1

Reviewer 1 Report

The article presents a good study on the application of conductive yarn to sensor design with different stitching patterns, however, there are some aspects to improve upon.

Fiber Material Properties

One key issue with the current study design is that the sensor performance of the conductive thread has not been included in the investigation. This makes it difficult to properly interpret the experimental results and the model for conductivity in the stitch design presented in the discussion.

Introduction

The introduction gives  a good overview of stitching research for sensors, but makes no mention of other possibilities such as piezoresistive/capacity designs and carbon-based sensors for wearable computing. It should at least be mentioned that other technologies are available to better contextualize why it is important to study silver-coated weavings, what are the advantages/disadvantages of using woven thread instead of carbon ink printing (for example).

Line 57-64: This is text from the MDPI template and must be removed before publication.

Line 106-113: Has different line spacing than the rest of the paper.

Table 2 specifics that two different threads were used in the sensor production, but it is not specified why a specific thread was used for the different stitch structures. Given that the threads very different linear resistance values (500 Ohm vs 50 Ohm) and diameters, it should be defined why each thread was used for which stitch type.

Line 81: It is stated that commercial conductive threads were used, but no information is given about the supplier. The commercial supplier must be stated. For example, see the experimental setup in Reference [2].

Line 159: The exact type of Instron machine must be defined.

Line 189: Error! Reference Source not found.

Line 229: Dimensions should be added to the image for a scale reference.

Line 230: Figures should be labeled with the Stitch Type for better readability.

Line 245 - 258: A model for conductivity is presented including the possibility that changes in dimensions of the conductive thread alters the local resistance of fiber sections. However, how would this actually occur (as stated in Line 48-51)? One can reasonably assume that conduction occurs through the silver particles, and the failure limit of silver is probably lower than the nylon, so one possibility is that cracks open up in the silver, leading to a decrease in conductivity (similar to the way that bend sensors work, https://pdfs.semanticscholar.org/0b20/9ec25ad2cff545130660b53b183923418343.pdf). However, since it is a woven thread, the application of tension could instead lead to an increase in surface contact between conductive surfaces as the cross-section of the thread decreases under tension, additionally, a better orientation along the direction of loading could occur. An answer to this one way or another would be easily supported if tensile testing had been conducted on the bare thread and not only on the stitched patterns, as it is the only way to separate one from the other.

Line 267-273: There is a discussion on how the increased cross-sectional area of conducting surfaces affects resistances of the stitch structures. Here a model could be presented which takes the linear resistance of the threads and the length of the thread used in each stitch structure, then compare it to the baseline undeformed resistance of the stitch structure. This would given an indication how many contact areas were formed in the production process, and how they theoretically open while under strain. This would give perspective on how to design stitch patterns in the future to achieve different performance characteristics (for example, maximizing sensor signal along the direction of strain, or perpendicular, or at another angle) and explain the current results.

Line 256: Error! Reference Source not found.

Line 309: It is stated that Type 304 2-ply gave the best performance with a gauge factor of 1.61, but there is no discussion of how this value compares to other strain sensors for wearable applications. This value is below that of nominal metal foil strain gauges, and the performance need highly depends on the final application (The abstract lists “monitoring such as body movement, heart monitoring, and limb articulation measurement.”). 1.61 may be perfect or inferior (for example it could be assumed heart monitoring relates to ECG electrodes where a very low gauge factor would probably be preferable), so more context is needed in the discussion, and to relate the results to past research. For example, both references [1] and [2] present results of woven sensors similar to Type 304 2-ply used in this study, and if you calculate the gauge factor from the results shown in Figure 5 (Type 1 curve) from [https://www.mdpi.com/1424-8220/14/3/4712/htm] the gauge factor appears to be 1.75, so the current results are comparable to past research. How are the current results building upon the results of other researchers, or does the present work mainly reinforce their results and show that the other stitch patterns are not ideal as strain sensors?

Author Response

Dear Reviewer,

I thank you for the time and effort for your comments. We have taken all of them on board and the manuscript has been revised accordingly. Please note that we have agreed to revise our manuscript to all of your comments and we clarified text, tables and figures. We also improved some of the English.

Our specific changes to the reviewer comments:

Your comment:

One key issue with the current study design is that the sensor performance of the conductive thread has not been included in the investigation. This makes it difficult to properly interpret the experimental results and the model for conductivity in the stitch design presented in the discussion.

Our answer:

The specification and the testing of the performance of the sewing threads has been done and inserted and p 87 to p 125. Agreed with the reviewer.

Your comments:

The introduction gives  a good overview of stitching research for sensors, but makes no mention of other possibilities such as piezoresistive/capacity designs and carbon-based sensors for wearable computing. It should at least be mentioned that other technologies are available to better contextualize why it is important to study silver-coated weavings, what are the advantages/disadvantages of using woven thread instead of carbon ink printing (for example).

Our Answer:

Piezoresistive/capacity designs inserted in Introduction and 3 references added: Agreed

Line 57-64: This is text from the MDPI template and must be removed before publication. Done

Line 106-113: Has different line spacing than the rest of the paper. Done

Table 2 specifics that two different threads were used in the sensor production, but it is not specified why a specific thread was used for the different stitch structures. Given that the threads very different linear resistance values (500 Ohm vs 50 Ohm) and diameters, it should be defined why each thread was used for which stitch type. Done

Line 81: It is stated that commercial conductive threads were used, but no information is given about the supplier. The commercial supplier must be stated. For example, see the experimental setup in Reference [2]. Done

Line 159: The exact type of Instron machine must be defined.Done

Line 189: Error! Reference Source not found. Done

Line 229: Dimensions should be added to the image for a scale reference. Done

Line 230: Figures should be labeled with the Stitch Type for better readability.Done

Line 245 - 258: A model for conductivity is presented including the possibility that changes in dimensions of the conductive thread alters the local resistance of fiber sections. However, how would this actually occur (as stated in Line 48-51)? One can reasonably assume that conduction occurs through the silver particles, and the failure limit of silver is probably lower than the nylon, so one possibility is that cracks open up in the silver, leading to a decrease in conductivity (similar to the way that bend sensors work, https://pdfs.semanticscholar.org/0b20/9ec25ad2cff545130660b53b183923418343.pdf). However, since it is a woven thread, the application of tension could instead lead to an increase in surface contact between conductive surfaces as the cross-section of the thread decreases under tension, additionally, a better orientation along the direction of loading could occur. An answer to this one way or another would be easily supported if tensile testing had been conducted on the bare thread and not only on the stitched patterns, as it is the only way to separate one from the other. Done line 317 to 322 added.

Line 267-273: There is a discussion on how the increased cross-sectional area of conducting surfaces affects resistances of the stitch structures. Here a model could be presented which takes the linear resistance of the threads and the length of the thread used in each stitch structure, then compare it to the baseline undeformed resistance of the stitch structure. This would given an indication how many contact areas were formed in the production process, and how they theoretically open while under strain. This would give perspective on how to design stitch patterns in the future to achieve different performance characteristics (for example, maximizing sensor signal along the direction of strain, or perpendicular, or at another angle) and explain the current results. Agree but not within the scope of this paper. We continue this path but due to loops and complex curvetures it is not realistic to do it without analysis of all structures used which takes specialist equipment and time. It will be in a forthcoming paper.

Line 256: Error! Reference Source not found.Done.

Line 309: It is stated that Type 304 2-ply gave the best performance with a gauge factor of 1.61, but there is no discussion of how this value compares to other strain sensors for wearable applications. This value is below that of nominal metal foil strain gauges, and the performance need highly depends on the final application (The abstract lists “monitoring such as body movement, heart monitoring, and limb articulation measurement.”). 1.61 may be perfect or inferior (for example it could be assumed heart monitoring relates to ECG electrodes where a very low gauge factor would probably be preferable), so more context is needed in the discussion, and to relate the results to past research. For example, both references [1] and [2] present results of woven sensors similar to Type 304 2-ply used in this study, and if you calculate the gauge factor from the results shown in Figure 5 (Type 1 curve) from [https://www.mdpi.com/1424-8220/14/3/4712/htm] the gauge factor appears to be 1.75, so the current results are comparable to past research. How are the current results building upon the results of other researchers, or does the present work mainly reinforce their results and show that the other stitch patterns are not ideal as strain sensors? Done, added comments in lines 377 to 379

Many thanks

G Stylios, on behalf of both authors

Reviewer 2 Report

In this

paper,

G.J. Mohr

report

s

on

the synthesis of

synthesis of 2

-

hydroxyethylsulfonyl

naphthalimide derivatives and

their

use as type fluorescent indicator dye

s

covalently and stably

linked to cellulose. In fact, the application of three different pH indicator dyes, with pKa ranging

from 3 to 7, allows in principle for covering a wider pH range

response od modified textiles

,

thus

opening the way to further potent

ial application and synthetic attempts.

T

he work seem to me

well organized

and self

-

consistent

, and

I think that this

paper does meet the

basic requirements of

Sensors and Actuators B

In this paper, G. Stylios reports on the investigation of seven different novel textile-based strain sensors and evaluates their performance, in terms of electrical resistance and mechanical properties.

Analysis of the effects of elongation with respect to resistance  indicated the ideal configuration for electrical properties. The sensor made with stitch type 304 2-ply was found to be the best performing in strain 308 sensing for wearable garment end uses.

The application of different types of textile stitch-based strain sensors, allows in principle for covering replace solid state sensors, thus opening the way to further potential application and synthetic attempts towards the development of  next generation of user-friendly interfaces on garments having the ability to provide physiological measurement to users.

The work seem to me well organized and self-consistent, and I think that this paper does meet the basic requirements of Materials.

Author Response

Dear Reviewer,

Thank you for your nice comments and encouragement. We have improved our manuscript and have revised it accordingly.

Many thanks

George Stylios, on behalf of both authors

Reviewer 3 Report

The authors report an investigation on textile-based strain sensors, evaluating their performance on the basis of the stitch used.

The topic is interesting and timely as the research on smart textiles is rapidly growing.

However the description of the techniques used needs to be improved by avoiding too basic and elementary concepts (e.g. the description of "linearity" lines 127-135) and possibly by stressing the peculiarities or difficulties encountered while applying these techniques to textile substrates/materials 

The results have to be presented with associated experimental errors, both in the text and in the tables. Without this information it is hard to support the discussion on the different performance.

Please add,to assess repeatability and stability:

- a graph similar to Figure 3 but with multiple runs  (not subsequent cycles) on the same sample (not averaged and with experaimtal errors) . 

- a graph comparing the simple one run output obtained from various but identical samples (304 2-ply would probably be the most relevant)

More details  (coating thickness, diameter, or other) should be reported on the 2-ply vs 4-ply conductive threads used, maybe reporting also images, to better support and discuss the drawn conclusions. Why is the 304 stitch with the 4-ply thread not reported?

Author Response

Dear Reviewer,

Thank you for your comments. We have made substantial changes to our manuscript to make it clearer and more accurate. 

Many thanks

George Stylios, on behalf of both authors.

However the description of the techniques used needs to be improved by avoiding too basic and elementary concepts (e.g. the description of "linearity" lines 127-135) and possibly by stressing the peculiarities or difficulties encountered while applying these techniques to textile substrates/materials This is done because some textile readers are not that familiar with those terms but more familiar with the techniques.

The results have to be presented with associated experimental errors, both in the text and in the tables. Without this information it is hard to support the discussion on the different performance.Done, see table 9 and 10 error drift.

Please add,to assess repeatability and stability:This has been accessed in the results given in tables 6 (b) and 8 (b) which have been revised to the comments of the two previous referees. 

- a graph similar to Figure 3 but with multiple runs  (not subsequent cycles) on the same sample (not averaged and with experaimtal errors) . Please see Tables 6 and 8 (b)

- a graph comparing the simple one run output obtained from various but identical samples (304 2-ply would probably be the most relevant)

More details  (coating thickness, diameter, or other) should be reported on the 2-ply vs 4-ply conductive threads used, maybe reporting also images, to better support and discuss the drawn conclusions. Why is the 304 stitch with the 4-ply thread not reported? Done, please see Table 3, figure 4.
